# Impact of Mygalin on Inflammatory Response Induced by Toll-like Receptor 2 Agonists and IFN-γ Activation

**DOI:** 10.3390/ijms251910555

**Published:** 2024-09-30

**Authors:** Nayara Del Santos, Ricardo Vázquez-Ramírez, Elizabeth Mendes, Pedro Ismael Silva Júnior, Monamaris Marques Borges

**Affiliations:** 1Bacteriology Laboratory, Butantan Institute, São Paulo 05585-000, Brazil; elizabeth.mendes@butantan.gov.br; 2Institute of Biomedical Research, National Autonomous University of Mexico, Mexico City 04510, Mexico; ricardo1v@yahoo.com; 3Laboratory for Applied Toxinology (LETA), Butantan Institute, São Paulo 05585-000, Brazil; pisjr@butantna.gov.br

**Keywords:** mygalin, acylpolyamine, cytokines, inflammation, TLR, molecular docking

## Abstract

Several natural products are being studied to identify new bioactive molecules with therapeutic potential for infections, immune modulation, and other pathologies. TLRs are a family of receptors that play a crucial role in the immune system, constituting the first line of immune defense. They recognize specific products derived from microorganisms that activate multiple pathways and transcription factors in target cells, which are vital for producing immune mediators. Mygalin is a synthetic acylpolyamine derived from hemocytes of the spider *Acanthoscurria gomesiana*. This molecule negatively regulates macrophage response to LPS stimulation by interacting with MD2 in the TLR4/MD2 complex. Here, we investigated the activity of Mygalin mediated by TLR2 agonists in cells treated with Pam3CSK4 (TLR2/1), Pam2CSK4, Zymosan (TLR2/6), and IFN-γ. Our data showed that Mygalin significantly inhibited stimulation with agonists and IFN-γ, reducing NO and IL-6 synthesis, regardless of the stimulation. There was also a significant reduction in the phosphorylation of proteins NF-κB p65 and STAT-1 in cells treated with Pam3CSK4. Molecular docking assays determined the molecular structure of Mygalin and agonists Pam3CSK4, Pam2CSK4, and Zymosan, as well as their interaction and free energy with the heterodimeric complexes TLR2/1 and TLR2/6. Mygalin interacted with the TLR1 and TLR2 dimer pathway through direct interaction with the agonists, and the ligand-binding domain was similar in both complexes. However, the binding of Mygalin was different from that of the agonists, since the interaction energy with the receptors was lower than with the agonists for their receptors. In conclusion, this study showed the great potential of Mygalin as a potent natural inhibitor of TLR2/1 and TLR2/6 and a suppressor of the inflammatory response induced by TLR2 agonists, in part due to its ability to interact with the heterodimeric complexes.

## 1. Introduction

Currently, there is a significant effort to define new compounds with pro- or anti-inflammatory activity that can be used as drugs to modulate the inflammatory response and control various pathologies in addition to infections. Compounds from diverse origins have been tested as agonists or antagonists in this process, aiming to control the synthesis and activation of multiple proteins and the emission of molecular signals that promote or inhibit the immune response and the control of diverse pathologies.

Innate immunity acts as the first line of defense, being characterized by a rapid response, independent of previous stimuli. It involves several closely integrated steps, constituted by different components, and can direct specific adaptive immunity [1].

The main receptors involved in the activation of the innate response in infections are Toll-like receptors (TLRs), which act as sensors of microorganism or nucleotide structures. They are transmembrane proteins, expressed on the cell surface (TLR1, 2, 4, 5, 6, and 10) or within intracellular compartments such as the endoplasmic reticulum, endosome, and lysosomes (TLR3, 7, 8, 9, 11, and 13). TLR4 is the only receptor expressed in both compartments [2]. Several chemical structures are recognized by TLRs. TLR2/1 and TLR2/6 receptors recognize bacteria-derived molecules, including lipoprotein, glycan peptide, lipoteichoic acid, and zymosan fungal wall, while TLR4 recognizes bacterial lipopolysaccharide (LPS) in addition to other molecules. Its recognition increases the expression of multiple genes and transduction signals involved in the inflammatory response, playing an important role in controlling infections [1,3].

Among all TLRs, TLR4 is the only one that requires the presence of the accessory molecule MD2 and forms an active heterodimer TLR4/MD2, which is essential for LPS recognition. Several studies are being carried out to identify small molecules to disrupt the interaction between TLR4/MD2 and suppress deleterious proinflammatory responses to LPS [4].

Briefly, the inflammatory response generated by TLR involves an orchestrated signal transduction pathway and requires the recruitment of intracellular adaptor molecules, among them MyD88 or TRIF. TLR activation initiates distinct signaling cascades through one or both main pathways: the MyD88-dependent pathway and the TRIF-dependent pathway. All TLRs except TLR3 recruit the MYD88-dependent pathway, while TLR4 uses both. TRIF-dependent signaling activates interferon regulatory factor (IRF) and secretion of type I IFNs, while MYD88-dependent pathways promote all TLR signaling with activation of nuclear factor NF-κB and mitogen-activated protein kinase (MAPK), and also the production of proinflammatory cytokines, IL-1, IL-6, IL-8, TNFα, chemokines, and others [5]. These signals are crucial for the initial phase of pathogen defense, allowing cells to regulate their antimicrobial and immune functions [6]. NF-κB plays a critical role in regulating the survival, activation and differentiation of innate immune cells and inflammatory T cells. Consequently, deregulated NF-κB activation contributes to the pathogenic processes of various inflammatory diseases. Dysfunction or dysregulation of TLR signaling contributes to the development of chronic inflammatory conditions associated with autoimmune, inflammatory, and infectious diseases [7].

TLR2 and TLR4 signaling use the same via MyD88-dependent pathway; however, unlike TLR4, TLR2 activation does not require the presence of the accessory molecule MD2 [2].

Pam3CSK4 and Pam2CSK4 are synthetic lipoprotein analogs, activators of the TLR2/TLR1 and TLR2/TLR6 heterodimer, which regulate the NFκB and MAPKs cascade and subsequently initiate a proinflammatory response by activating target genes related to cytokine expression and production of reactive oxygen and nitrogen [8,9].

Macrophages are essential components of innate immunity, have diversity and plasticity, and multiple functions in inhibition or promotion of cell proliferation and tissue repair. Two populations of macrophages, called M1 and M2, have been described. They differ in phenotype, activation status, and immune function. The M1 population is primarily activated by IFN-γ and microbial products via TLRs. These cells are highly phagocytic, release pro-inflammatory cytokines, and have a marked glycolytic metabolism. They are microbicidal, tumoricidal, and promote CD4+ Th1 response. The M2 population emits anti-inflammatory signals, producing anti-inflammatory cytokines, tissue repair, immune tolerance and tumor progression, and promoting CD4+ Th2 response. This population undergoes fatty acid metabolism and mitochondrial oxidative phosphorylation. The imbalance of these populations may be associated with various diseases or chronic inflammation [10,11,12]. This dual mode of macrophage activity functions in the establishment, as well as the resolution or prevention of, inflammation.

The recognition and activation of macrophages via TLRs agonists and IFN-γ may synergize and induce transcriptional, post-transcriptional, and epigenetic factor signaling regulatory networks such as NF-κB, AP-1, and JAK/STAT signaling pathways, as well as MAPkinase activation [13].

Stimulation of macrophages with IFN-γ promotes a remodeling of the metabolic and epigenetic functions of these cells. This induces the expression of transcription factors such as STATs (transcription signal and activator) and IRF (Interferon regulatory factor), proteins which are central pathways in modulating macrophage polarization. The treatment of macrophages with IFN-γ following TLR stimulation has a synergic effect on the induction of Stat1/IRF1-dependent signals, which is a site of convergence between IFN-γ and TLR signaling pathways. Such synergy seems to be a mechanism for integrating various molecular signs to allow multiple integrative signals for coordination of gene expression during inflammatory response in various forms of cellular activation [14].

Pharmacological interventions that modulate the activation of Toll-like receptors have received considerable attention, in this regard. Among these molecules, polyamines (putrescine, spermidine, and spermine) are small aliphatic amines present in all living organisms and have broad biological functions. These molecules participate in the control of cell proliferation and differentiation, regulation of protein synthesis, and gene expression, as they can interact with DNA and RNA segments [15], promoting conformational changes in the structure and function of these molecules [16]. Polyamines also modulate intracellular signals [17] and immune functions, depending on their nature [18,19].

Mygalin is a synthetic acylpolyamine analog of spermidine, originally isolated from hemocytes of the spider *Acanthoscurria gomesiana*, with a low molecular weight (417 Da), identified as N1, N8-bis(2,5-dihydroxybenzoyl) bis-acylpolyamine [20]. This molecule is not cytotoxic, reduces the inflammatory LPS activity by interfering with the TLR4/MD2 complex, and binds to enzymes such as cyclooxygenase (COX-2) and 5-lipoxygenase (5-LOX) [21], which participate in the control of the inflammatory response.

This study contributes to defining how Mygalin can directly regulate the activation and effector functions of macrophages induced by different Toll-like receptor 2 agonists, identify molecular mechanisms of action of this molecule, and explore potential applications.

## 2. Results

### 2.1. Influence of Mygalin in the Response Induced by TLR2 Agonists

The main function of TLRs in infections is the recognition of pathogens and the induction of innate immunity through antimicrobial activities and the production of immune mediators. This interaction regulates the intracellular signaling cascade of molecules, which depends on the nature of the stimulus, the type of TLRs activated, and the adaptor molecules [22]. Previous studies have shown that Mygalin negatively regulates the innate response mediated by LPS by interfering with TLR4/MD2. Here, other independent MD2 mechanisms will be analyzed via TLR2 activation.

#### 2.1.1. Effect of Pre-Treatment with Mygalin on Pam3CSK4 (TLR2/1-Induced Response)

Figure 1 shows that pretreatment with Mygalin inhibited the synthesis of NO, regardless of the dose, while only treatment with the highest dose of this compound reduced the presence of IL-6, suggesting a dose-dependent effect. No interference in TNF-α synthesis was observed. These results suggest that Mygalin can modulate the inflammatory response via TLR2/1, as with what is described with LPS, but differing in TNF-α synthesis, where Pam3CSK4 did not affect the presence of this mediator. This indicates that Mygalin may influence TLR activation differently, depending on the nature of the stimulus used.

#### 2.1.2. Effect on TLR2/6 Agonists

Here, we investigate whether the activity of Mygalin is mediated by TLR2/6, using two specific agonists. Pam2CSK4, a synthetic diacylated lipopeptide from bacteria, and Zymosan, which comes from yeast cell walls and consists of protein–carbohydrate complexes. These agonists were chosen due to their differences in origin and structure

##### Pam2CSK4-Induced Response

We next investigated the effect of Mygalin on the production of NO and IL-6 in Pam2CSK4-treated J774 cells. As shown in Figure 2, Mygalin inhibited the production of NO, regardless of the dose tested. However, IL-6 synthesis was reduced only during treatment with the highest dose of the compound. Lipoproteins (LPs) are components of Gram-positive and Gram-negative bacteria, and mimic the acylated amino terminus groups [8]. The data obtained showed that the effect of Mygalin on the cell response induced by treatment with the TLR2/1 ligand, tri-acylated synthetic lipoprotein (Pam3CSK4), and the TLR2/6 ligand, di-acylated synthetic lipopeptide (Pam2CSK4), was similar despite the difference in the number of acyl groups in the molecules.

##### Zymosan-Induced Response

TLR2 plays a role in detecting various molecular patterns associated with pathogens from different microorganisms. Zymosan is a TLR2/6 ligand derived from the yeast cell wall, and consists of carbohydrate–protein complexes. We investigated the effect of Mygalin on the inflammatory response induced by this agonist. Figure 3 shows that Zymosan induced the synthesis of all mediators analyzed. Pretreatment of cells with Mygalin reduced NO synthesis, regardless of the dose used, while IL-6 was reduced only when the cells were pretreated with the highest dose of the compound, 360 µM. However, this treatment did not affect TNF-α production. This anti-inflammatory effect was like that observed during stimulation of cells with Pam3CSK4 and Pam2CSK4, which are bacterial lipoproteins. These data suggest that Mygalin acts similarly in the inflammatory response induced by agonists of TLR2/1 and TLR2/6, independent of the nature and origin of the agonist.

#### 2.1.3. Effect of Mygalin Pre-Treatment on IFN-γ Activation

IFN-γ is a crucial pro-inflammatory cytokine in both innate and adaptive immune responses, particularly in macrophage activation in response to pathogens. The presence of this cytokine impacts several immune functions, such as pathogen survival, antigen recognition and presentation, and the induction of antiviral states and microbicidal functions. Additionally, IFN-γ can enhance cellular responses to TLR ligands. Due to the intense inflammatory and immunomodulatory characteristics of IFN-γ, we analyzed the effect of Mygalin on macrophage activation by this cytokine, with or without the agonist Pam3CSK4. Figure 4 shows that pre-treatment of J774 cells with Mygalin significantly reduced NO synthesis in IFN-γ-activated macrophages. However, Mygalin did not affect the levels of IL-6 and TNF-α induced by this cytokine, whereas LPS, used as a positive control, significantly increased the production of these cytokines.

#### 2.1.4. Influence of the Pre-Treatment of Cells with Mygalin during the Activation with IFN-γ plus Pam3CSK

Multiple signals derived from host factors and pathogen products are required to regulate the functions of macrophages during the innate immune response. The synergistic effects between IFN-γ and macrophage activation during the recognition of pathogens via TLR may enhance the inflammatory reaction by interfering with the expression of various inflammatory response genes.

Several studies have demonstrated the synergistic effects of IFN-γ and TLR2. IFN-γ modulates TLR2-induced signaling by suppressing MAPK activation [23] and recruits and activates immune cells in the tumor microenvironment [24]. Furthermore, cells overexpressing TLR2 that were activated by BCG responded similarly to those activated with Pam3CSK4 [25], while blocking TLR2 reduced IFN-γ production in response to BCG in human dendritic cells [26].

Therefore, tests were performed to understand the influence of Mygalin on activation by the combination of IFN-γ and Pam3CSK4. Figure 5 shows that this combination resulted in elevated NO synthesis compared to unstimulated cells, and pre-treatment with Mygalin reduced the levels of this mediator only in the groups treated with the higher dose (360 µM) of the compound, suggesting a dose-dependent effect. The combination of IFN-γ and Pam3CSK4 also increased the levels of IL-6 and TNF-α. There was a slight but significant reduction in IL-6 synthesis after treatment with the higher dose of Mygalin (360 µM), and levels of TNF-α remained unchanged.

### 2.2. Pre-Activation of Macrophages with Pam3CSK4 or IFN-γ plus Pam3CSK4 and Treatment with Mygalin

The combinations of TLR agonists and interferons have synergistic effects on the production of pro-inflammatory cytokines and nitric oxide (NO) in macrophages [27]. Excessive production of inflammatory cytokines can lead to tissue destruction and be toxic to the host. In another protocol, cells were initially activated with IFN-γ and Pam3CSK4 and subsequently treated with Mygalin to mimic the effect on an already established inflammatory response and to confirm whether the same level of inhibition observed previously would be achieved.

#### 2.2.1. Effect of Mygalin on Macrophages Pre-Activated with Pam3CSK4

Figure 6 shows that the effect of Mygalin on cells preactivated with the TLR2/1 (Pam3CSK4) agonist resulted in a less pronounced reduction in NO in a dose-dependent manner, compared to the protocol where cells were first treated with Mygalin followed by activation with the agonist. A similar pattern was observed for the synthesis of IL-6 and TNF-α. This highlights the fact that the timing of Mygalin exposure influences the observed effect and confirms that pre-treatment with Mygalin before TLR2 activation by the agonist has a more pronounced suppressive effect on the innate response.

#### 2.2.2. Effect of Mygalin on Macrophages Pre-Activated with IFN-γ plus Pam3CSK4

As shown in Figure 7, pre-activation of J774 cells with Pam3CSK4 plus IFN-γ reduced the synthesis of NO and IL-6 in a dose-dependent manner, without altering TNF-α levels. These results were like those observed with Pam3CSK4 alone, but the reduction effect was less pronounced.

### 2.3. Mygalin Suppresses the Inflammatory Response Induced by Pam3CSK4 by Interfering with NF-κB p65 and STAT-1 Activation

Targeting TLR signaling represents a new challenge for the treatment of many infections and diseases. Recognition of different pathogen-associated molecules by TLRs leads to the activation of transcription factors such as NF-κB, AP-1, IRF3, and STAT1, which are crucial for cytokine synthesis and the establishment of the inflammatory response [28]. Since Mygalin can suppress cytokine production, depending on the nature of the stimulus, we decided to evaluate its effect on the expression of proteins involved in molecular signal transduction following activation with the Pam3CSK4 agonist, to better elucidate the molecular mechanism of action of Mygalin. Pam3CSK4 was used as a TLR2/1 activator known to signal via MyD88 and activate the phosphorylation of proteins involved in NF-κB and STAT1 activation. Figure 8 shows that stimulation of cells with Pam3CSK4 (300 ng/mL) induced the phosphorylation of NF-κB p65 and STAT1. The addition of Mygalin (90 and 360 μM) to this group significantly reduced the expression of these proteins, with the most pronounced reduction observed with the highest dose (360 μM). LPS, used as a positive control, promoted the phosphorylation of both transcription factors.

### 2.4. In Silico Analysis of Mygalin Interaction with TLR2/TLR1 and TLR2/TLR6

The use of molecular modeling significantly contributes to the optimization of bioactive molecules and is a widely used strategy in the development of new compounds with potential applications. Molecular docking is based on the binding affinity between a potential binder and a molecular target, which can be a receptor, enzyme, or other molecule. This analysis explores the interaction conformations between a ligand and its molecular targets, as well as estimating the free energy of association with the receptor—critical aspects during intermolecular recognition. Therefore, the initial analyses focused on the molecular structure of Mygalin, TLR2, Pam3CSK4, Pam2CSK4 agonists, and Zymosan, along with the complex interactions between Mygalin and Toll-Like Receptors 2/1 and 2/6, including the free energy assessment of these interactions.

#### 2.4.1. Molecular Structure of Ligands and TLR2/TLR1

The electronic structure and physicochemical properties of Mygalin, Pam3CSK4, Pam2CSK4, and Zymosan were obtained through ab initio calculations. The ligands exhibited distinct patterns of electrostatic potential (EP) and dipole moments (Figure 9). Mygalin and Zymosan have a neutral net charge, with heterogeneous positive and negative EP patterns along the molecules derived from polar groups (OH and NH) and hydrophobic groups (aromatic rings and methyl groups). In contrast, Pam3CSK4 and Pam2CSK4 show EP with two well-defined phases: a positive-polar (blue) phase in the peptide region and a neutral-hydrophobic (green) phase, due to the triacyls (two ester-bound lipids and one amide-bound lipid) in Pam3CSK4, and diacyls in Pam2CSK4. The dipole moments and polar surface area (PSA) confirm the high polarity of all the ligands. Mygalin and Zymosan exhibit lower dipole moments compared to the other ligands due to their neutral nature, with Mygalin’s reduced dipole moment attributed to its molecular symmetry (Figure 9 and Table 1). Mygalin is the least polar of the four ligands, but has good hydrogen bonding potential. Zymosan is more polar than Mygalin, as indicated by its dipole moment, and also has the highest PSA among the ligands, due to its large number of hydroxyl groups capable of forming hydrogen bonds. However, Zymosan is reported to be a water-insoluble glycoside [29]. Pam3CSK4 and Pam2CSK4 exhibit biphasic EP, due to an electronic imbalance caused by the positive net charges (cationic) in the peptide region, contrasting with neutral regions due to lipid chains, resulting in very large dipoles. The PSA is smaller, due to the contribution of the hydrophobic lipid chains. Under biological conditions (pH 7.2), these agonists are protonated, due to the ammonium groups (NH3+). The lysine side chains in the peptide regions of Pam3CSK4 and Pam2CSK4 contain four ε-ammonium groups. In Pam2CSK4, the additional ammonium group was considered neutral because the protonated configuration was highly unfavorable in terms of interaction energy with TLR6.

Regarding receptors, the TLR2/1 complex, in the absence of a ligand, has two hydrophobic pockets: one in TLR2, with a volume of 912 Å^3^, and a smaller one in TLR1, with a volume of 255 Å^3^ (Figure 10). In contrast, the TLR2/6 complex has only one hydrophobic binding pocket in TLR2, with a volume of 1175 Å^3^ (Table 1 and Table 2)).

#### 2.4.2. Ligand–Receptor Complexes

The X-ray structures of the Pam3CSK4, Pam2CSK4, and Zymosan docking-analysis complexes were used to compare the results of the docking analysis of Mygalin docked to TLR2/1 and TLR2/6 heterodimers. Residues in TLR2/1 and TLR2/6 bound to ligands, showing hydrophilic (polar) interactions, reached approximately 3.4 Å, while those with hydrophobic (van der Waals) interactions reached 4.5 Å (Figure 11). The number of binding residues was identified, with most interactions being hydrophobic compared to polar, and only 1 (in Pam3CSK4) and 2 (in Pam2CSK4) being ionic (Appendix A).

The interaction of Pam3CSK4 with TLR2/1 involved the occupancy of two hydrophobic binding pockets, one on TLR2 and one on TLR1. The two ester-linked lipid chains of Pam3CSK4 occupied TLR2, while the amide-linked lipid chain occupied TLR1, resulting in abundant hydrophobic interactions (Appendix A), as previously reported [8]. The peptide region of Pam3CSK4 remained mostly outside the hydrophobic pockets, due to the polar groups on the side chains of the residues, establishing hydrogen bonds and ionic interactions in the solvent-exposed regions of both receptors (Figure 11).

In TLR2/6, Pam2CSK4 and Zymosan occupied the single binding pocket in TLR2. The two ester-linked lipid chains of Pam2CSK4 and the monosaccharides 1–6 of Zymosan occupied the hydrophobic pocket region of TLR2 through abundant hydrophobic interactions (Figure 12 and Appendix A). Similar to TLR2/1, the peptide region of Pam2CSK4 in TLR2/6 remained mostly outside the hydrophobic pocket, establishing hydrogen bonds and ionic interactions in the solvent-exposed regions of TLR2 and TLR6 (Figure 12). In the case of Zymosan, monosaccharides 5–7 were bound to TLR2/6 residues exposed to the solvent, while monosaccharides 8–10 remained free.

The interactions of Mygalin with TLR2/1 and TLR2/6 were similar, particularly in its interaction with TLR2. Mygalin interacted with 11 residues of TLR2 in both heterodimers, with 7 conserved residues identified (F325, Y326, V348, F349, L350, and P352). Mygalin docked at the entrance of the binding pocket and did not occupy the bottom of the hydrophobic pocket in TLR2 (Figure 11 and Figure 12).

#### 2.4.3. Free Energy of Interaction of Ligands with TLR2/1 and TLR2/6 Dimers

The interaction energies of Mygalin, Pam3CSK4, Pam2CSK4, and Zymosan (Figure 13) with their respective receptors, were calculated based on quantum mechanics theory. Crystallographic complexes of Pam3CSK4 with TLR2/1 and Pam2CSK4 with TLR2/6 were analyzed, as well as the docked complexes of Mygalin and Zymosan with the same receptors.

In the TLR2/1–ligand complexes, Pam3CSK4 exhibited the most favorable free-energy interaction (−1352.26 kJ/mol) compared to Mygalin (−547.58 kJ/mol). Although Mygalin’s interaction energy indicates good affinity for TLR2/1, Pam3CSK4 shows a significantly stronger interaction. Hydrophobic interactions are predominant in the TLR2/1-ligand complex; however, polar interactions also contribute substantially to the stabilization of both ligands.

For the TLR2/6–ligand complex, Pam2CSK4 shows the most favorable free energy of interaction (−1352.26 kJ/mol), due to the highest number of interactions and the contribution of two ionic interactions. Zymosan, despite having abundant interactions, lacks ionic interactions, resulting in an intermediate free-energy interaction(−695.52 kJ/mol). In contrast, the TLR2/6–Mygalin complex, with the fewest interactions and no ionic interactions, has the lowest free energy (−385.09 kJ/mol) among the three ligands analyzed.

When comparing Mygalin complexes, the TLR2/1–Mygalin complex shows a better free energy, which is attributed to a higher number of interactions, particularly hydrogen bonds. The TLR2/1–Mygalin complex features eight hydrogen bonds, compared to only four in the TLR2/6–Mygalin complex, which may account for the superior free energy observed in the TLR2/1–Mygalin interaction.

**Figure 13 ijms-25-10555-f013:**
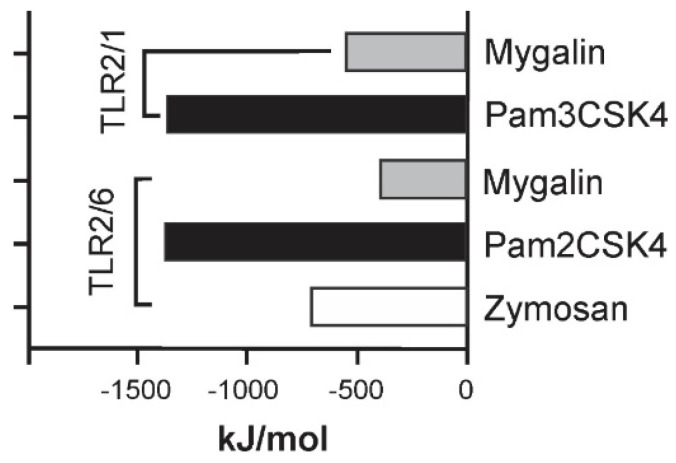
Comparison of interaction energies between agonists and Mygalin with TLR2/1 and TLR2/6.

## 3. Discussion

Research and development of new drugs derived from natural products involve the identification, preparation, and characterization of the mechanisms of action of biologically active compounds. These compounds are essential in the final drug development process. The vast chemical diversity of natural products allows for the discovery of a wide range of bioactive molecules that can be explored as promising new therapeutic options. Natural polyamines, characterized by multiple amine groups, play various roles in biological processes, including the regulation of cell growth, protein expression, and modulation of immune response. Their synthetic analogs have been studied as potential therapeutic agents, due to their pharmacological properties such as selectively inhibiting certain metabolic pathways related to tumor growth or modulating inflammatory processes [30].

Recently, through molecular docking, we demonstrated that Mygalin can influence the regulation of the innate immune response by interacting with the TLR4/MD2 complex, reducing the production of pro-inflammatory mediators induced by LPS, including TNF-α and IL-6 cytokine synthesis, as well as mRNA expression for COX-2, iNOS, and NF-κB [31]. Due to the complexity of the inflammatory response and existing gaps in understanding various aspects of Mygalin’s mechanism of action, our study aims to elucidate how it acts in response to different stimuli, to demonstrate its pharmacological potential for therapeutic applications. Additionally, cytokine production and other molecular signals are important mechanisms in the screening process of newly isolated molecules. In the present study, we expanded these findings by investigating the mechanisms of action of Mygalin on the innate immune response in vitro, using murine macrophage J774A.1 cell lines, stimulated by TLR2/1 or TLR2/6 agonists, either alone or in combination with IFN-γ.

We demonstrated that pre-treatment with Mygalin and subsequent activation with TLR2/1 (Pam3CSK4) and TLR2/6 (Pam2CSK4 and Zymosan) agonists promoted a marked reduction in NO and IL-6 synthesis, without interfering with TNF-α production. The most significant effects were observed in NO production, where there was a pronounced reduction, regardless of the dose used. Our data suggest that Mygalin may act via TLR2/1 and TLR2/6, blocking the inflammatory activity generated by Pam3CSK4, Pam2CSK4, and Zymosan activation. These data were similar to those for the inflammation induced by LPS via TLR4/MD2 [31], demonstrating that Mygalin’s action is independent of the presence of the accessory molecule MD2, which is exclusively associated with the activation of TLR4.

IFN-γ is a key factor in the development of cellular immunity, playing a critical role in the recognition and elimination of pathogens. It is initially produced by natural killer cells and innate lymphoid cells (ILCs), and later by CD4+ Th1 cells and CD8+ cytotoxic T lymphocytes. The activation of TLR by agonists and IFN-γ may synergize and induce signaling regulatory networks such as NF-κB, AP-1, and JAK/STAT signaling pathways, and MAP kinase activation in macrophages [13]. When IFN-γ recognizes its receptor, it emits molecular signals that regulate the pro-inflammatory functions of macrophages and can synergize with TLRs. These immunomodulatory effects enhance antigen processing and presentation, and the production of reactive oxygen species (ROS) and reactive nitrogen intermediates (RNIs), which have antimicrobial functions, as well as contributing to cell proliferation, apoptosis, and responses against various pathogens [32].

Thus, assays were performed using macrophages pre-treated with Mygalin and subsequently activated with IFN-γ, either associated with TLR2 agonists (Pam3CSK4) or not. It was shown that the addition of Mygalin to macrophages activated with IFN-γ drastically reduced NO production, without altering IL-6 and TNF-α cytokine production. In assays where cells were pre-treated with Mygalin and subsequently activated with Pam3CSK4 + IFN-γ, there was a significant reduction in NO production in a dose-dependent manner, and a less pronounced reduction in IL-6 and TNF-α, unlike the data obtained with TLR2 agonists in the absence of cytokine, where there was also an intense reduction in IL-6. Other compounds, such as CU-CPT22, the first competitive antagonist described with high selectivity for TLR2/1 heterodimers in mice, showed activity similar to Mygalin, inhibiting TNF-α, IL-1β, and iNOS production [33]. These data suggest that Mygalin, in addition to controlling the response to microbial components represented by Toll-like receptor agonists, can influence the inflammatory response generated by IFN-γ stimulation, reducing NO synthesis.

To mimic the action of Mygalin on a previously established inflammatory response, protocol B was performed, in which cells were first activated with Pam3CSK4 associated or not with IFN-γ, and subsequently treated with Mygalin. We observed a significant reduction in Mygalin’s ability to suppress immune mediator production with this protocol, which was considerably less than that obtained in protocol A, where cells were pre-treated with Mygalin before agonist addition. This suggests that the timing of exposure to the molecule interferes with its anti-inflammatory action, and the presence of IFN-γ may attenuate this effect. The reduction of pro-inflammatory cytokines by blocking TLR2 may be a strategy in the treatment of inflammatory diseases, especially those with excessive production of inflammatory cytokines such as TNF-α, IL-1, and IL-6 [34,35].

The inflammatory response is characterized by the modulation of various signaling molecules and transcription factors such as NF-κB, AP-1, and STAT-1, which control macrophage activation and occur in response to TLR and IFN-γ activation, characteristic of M1 phenotype macrophages. Pam3CSK4 binding to TLR2/1 activates signaling pathways through the translocation of transcription factors such as NF-κB [36]. Therefore, assays were conducted to understand the effect of Mygalin on modulating these pathways. Pre-treatment of macrophages with Mygalin (360 μM), followed by activation with Pam3CSK4, resulted in the inhibition of NF-κB p65 activation, suggesting that Mygalin-induced cytokine suppression may involve gene regulation through NF-κB activation, preventing its translocation and consequently inhibiting the inflammatory cascade induced by Pam3CSK4. This effect is consistent with a previous study by Jong-Bin Kim and colleagues [36] on Raw 264.7 cells, where treatment with Poncirin, a flavanone glycoside with anti-inflammatory properties, inhibited PGE2 and IL-6 production. It was demonstrated that Poncirin reduced iNOS, COX-2, TNF-α, and IL-6 expression, suggesting that its anti-inflammatory properties may be attributed to negative regulation of NF-κB binding activity.

Macrophage polarization requires transcription signals that are associated with different phenotypes. IRF/STAT signaling is a central pathway in the modulation of macrophage polarization. A predominance of IFN-γ, NF-κB, and IRF/STAT1 activation promotes M1 polarization. In contrast, a predominance of IRF/STAT3 and STAT6 activation by IL-4/13 and IL-10 increases M2 macrophages [13,14].

TLR2 plays an important role in this regulation and the control of several inflammatory diseases, stimulating multiple signal transduction pathways that are MyD88-dependent and TRIF-independent [37]. It has been suggested that multiple TLRs (2, 4, and 9) induce the phosphorylation of STAT1 in a time-dependent manner. Additionally, TLR2 and TLR4, associated with the plasma membrane, induce phosphorylation faster than endosomal TLRs, due to intracellular trafficking [38]. The effect of Mygalin on the activation of TLR2 and TLR4 confirmed the anti-inflammatory effect of this molecule, regardless of whether a TLR2 or TLR4 agonist was used. This is likely because both receptors use the MyD88-dependent pathway, which controls the expression of pro-inflammatory molecules mediated by the transcription factor NF-κB. However, TLR2 does not have the MD2 molecule of the TLR4/MD2 complex.

We also demonstrated that Mygalin reduced STAT-1 phosphorylation in macrophages activated with Pam3CSK4. This suggests a direct interference with TLR2 activation, confirming the information obtained in the in silico analyses.

In silico analysis of the interaction potential of Mygalin in the heterodimeric complexes TLR2/1 and TLR2/6 shows that Mygalin can stably bind the ligand-binding domain similarly in both complexes. However, Mygalin’s binding is different from that of the agonists Pam3CSK4 (TLR2/1), Pam2CSK4, and Zymosan (TLR2/6), which deeply occupy the binding pocket. The volume of the agonists and binding pockets in the receptors also indicates complete occupancy (Table 1). All three agonists bind to the hydrophobic pockets, as well as to the solvent-exposed region at the interface between the heterodimers (TLR2-TLR1 and TLR2-TLR6). Meanwhile, Mygalin only binds to the neck of the binding pocket (the boundary between both monomers), and cannot deeply associate with the fully hydrophobic binding pocket in both heterodimers. Therefore, Mygalin establishes stable complexes located at a transition site, at the interface between the dimers, which resembles the transition groups from the hydrophobic to the polar region of the agonists (Pam3CSK4 and Pam2CSK4).

The interaction free energies of Pam3CSK4, Pam2CSK4, and Zymosan were higher than those of Mygalin in TLR2/1 and TLR2/6 (Figure 13), indicating a higher affinity of the agonists for their receptors. In the TLR2 activation process, some molecular mechanisms of the TLRs’ interaction have been reported that could help explain the in vitro results of this work. Dimeric arrangements are found in all known TLRs, and dimerization of extracellular domains brings intracellular TIR domains closer, promoting their juxtaposition and initiating the signaling cascade [8,39,40,41]. The crystallographic structure of the unliganded dimer (inactive) of TLR8 shows conformational differences compared to the liganded dimer (active), and is considered a resting state [40,42]. Unliganded dimers (inactive) have also been reported in TLR9 [43], and are described as “open” conformations due to the angle formed by the two monomers, in contrast to “closed” (active) conformations [42]. Thus, “accidental” signaling cascades due to the “collision” of two receptor monomers are avoided. Ligand binding is considered a requirement for conformational changes to occur in the receptor and promote activation [43].

Mygalin in TLR2/1 and TLR2/6 complexes partially occupies the ligand-binding domain, and the binding energies represent the lowest affinity for the receptors. This could mean that the interaction energy is sufficient to stabilize the heterodimers and prevent conformational changes that promote signaling. It can be proposed that Mygalin binds to a preformed resting-state dimer, maintaining the receptors in that state, similar to TLR8. Small molecule inhibitors stabilize dimers in the resting state and prevent conformational changes required for activation, as reported for TLR8 [40,41].

TLR2 plays important roles in the innate immune system, and is related to many inflammatory diseases. In TLR signaling, the presence of negative regulatory molecules may function to prevent binding to the ligand receptor, degrade target proteins, or interfere with intracellular signaling pathways to inhibit signal transduction or transcription responsible for regulating inflammation under various conditions [44]. Therefore, the discovery of a low-cytotoxicity TLR2 antagonist, such as Mygalin, holds promise for the treatment of many inflammatory diseases. Studies on its potential application can help elucidate mechanisms for inhibiting excessive activation of TLRs in various infections and pathologies.

## 4. Materials and Methods

### 4.1. In Vitro Studies

#### 4.1.1. Cell Culture

The murine macrophage cell line J77A.1 was cultivated in complete RPMI (Gibco, Invitrogen Corporation, Waltham, MA, USA), supplemented with bovine fetal serum (Gibco Invitrogen Corporation) to 10% (FBS) and 25 µg/mL of gentamycin. The cells were then seeded at a density of 1 × 10^6^/500 µL in 48 well plates (Corning, New York, NY, USA) kept overnight in RPMI 1640, at 37 °C in humidified 5% CO_2_ atmosphere. The next day, the cells were washed, cultured in a new RPMI medium, and stimulated according to the protocol established for the assay.

#### 4.1.2. Treatments of the Cells

J774A.1 cells, after overnight culture, were treated according to two protocols. In the protocol A, adherent cells were pretreated with Mygalin for 1 h and later activated with TLR 2/1 agonists, Pam3CSK4 (300 ng/mL), TLR 2/6, Pam2CSK4 (300 ng/mL), Zymosan (10 µg/mL) or IFN-γ (10 ng/mL); all agonists were from InvivoGen, San Diego, USA. In another protocol (B), the cells were initially pre-activated with the TLR 2/1 receptor agonists, Pam3CSK4 (300 ng/mL), in the presence or absence of IFN-γ (10 ng/mL) for 20–24 h and later treated with Mygalin (90 and 360 µm). LPS (100 ng/mL) was used as a control in all tests. After 20–24 h of stimulus, with different protocols, the supernatants were obtained for quantification of nitric oxide dosage (NO) and cytokines (IL-6 and TNF-α).

#### 4.1.3. Measurement of NO and Inflammatory Cytokines

The Nitrite accumulation was measured in culture supernatants using the Griess reaction [45]. Briefly, for nitrite dosage (NO_2_^−^), 50 µL of the supernatant and 50 µL of 1% sulfanilamide solution were added to a 96-well plate and incubated for 10 min at room temperature. Then, 50 µL of NED to 0.1%was added, followed by additional incubation under the same conditions. Absorbance was measured at 550 nm. The concentration of NO_2_^−^ in the sample was based on the standard sodium nitrite curve (1–100 µM).

The presence of IL-6 and TNF-α was analyzed in the supernatants of the cell cultures using ELISA immunoenzymatic assay kits (cat. no. 88-7064-22, 88-7324-88; Invitrogen, Thermo Fisher Scientific, Inc., Waltham, MA, USA), following the manufacturer’s guidelines. The reaction was measured at 450 nm in a microplate reader (Multiskan EX, Primary EIA, Thermo Fisher Scientific, Inc., Waltham, MA, USA). The amount of cytokines present in the samples was calculated based on standard curves. All tests were performed in triplicate.

#### 4.1.4. Western Blot Analysis

Cells were pretreated with Mygalin for 1 h and later activated with Pam3CSK4 (300 ng/mL) by 6 h. Next, cells were harvested and proteins extracted at 4 °C using RIPA buffer (NaCl 150 mm, Triton X-100 to 1%, 0.5%deoxicolato sodium, SDS at 0.1%, Tris-HCl 50 mm pH 8, protease inhibitor). The lysed cells were centrifuged at 14,000 rpm by 20 min to 4 °C and protein concentration was determined by the Pierce ™ BCA (Thermo Scientific, Inc., Waltham, MA, USA) protein kit. Proteins (30–40 ug/mL) were separated by SDS-PAGE on a 10% polyacrylamide gel and subsequently transferred to the nitrocellulose membrane. Blots were incubated with the appropriated primary antibody anti- STAT-1, anti-β-actina or anti-NF-κB/p65 at 1:1500 (Cell Signalling Inc, Denver, MA, USA). IgG antibody anti-rabbit conjugated with peroxidase was used as a secondary antibody at 1:2000 dilution. Protein bands were detected by refined Signalfire ™ ECL and the images were captured and analyzed by electronic documentation system (Uvitec, Cambridge, UK).

#### 4.1.5. Statistical Analysis

Data are reported as the mean ± SEM of at least 3 independent experiments, each with at least 3 independent observations. Statistical evaluation was performed using the Student’s *t*-test, where the values *p* < 0.05 were considered significant, analyzed by the Prism 8 ^®^ GraphPad Program (Graph Pad, San Diego, CA, USA).

### 4.2. In Silico Studies

#### 4.2.1. Molecular Structure of TLR2/TLR1 and TLR2/TLR6 Complexes and Ligands

The crystal structures of the heterodimeric proteins for in silico analysis were obtained from the Protein Data Bank [46], using the PDB ID: 2Z7X (TLR2/TLR1) and 3A79 (TLR2/TLR6). The volume of the binding pocket cavities of each heterodimer was calculated using the Swiss-PDB Viewer program [47].

The molecular structure of Mygalin (C_21_H_27_N_3_O_6_) was retrieved from the ZINC database (ZINC ID: 87529034) [48]. Zymosan is a β-1,3-glucan polysaccharide; it has a large and complex branched structure with a molecular weight of approximately 296 kDa, and is usually fragmented into small chains [29]. Therefore, a ten-monosaccharide Zymosan fragment was modeled using Spartan’20 [Wavefunction Inc., Irvine, CA, USA, 2020] from a chain of three β-1,3-glucopyranoses, retrieved from ChemSpider ID: 9550471 [49]. While Pam3CSK4 (C_81_H_156_N_10_O_13_S) and Pam2CSK4 (C_65_H_126_N_10_O_12_S) were extracted from co-crystallizations with TLR2/TLR1 (PDB ID: 2Z7X) and TLR2/TLR6 (PDB ID: 3A79), respectively, using the Discovery Studio Visualizer (DSV) from Dassault Systèmes BIOVIA, San Diego, USA. Before docking analysis, geometry and energy-profile optimization calculations were carried out to obtain the lowest energy conformations of Mygalin and Zymosan. Finally, some electronic and physicochemical properties of the four ligands were calculated in the conformations established in the complexes with the receptors, including total energy, dipole moment, volume, polar surface area, and electrostatic potential (EP).

#### 4.2.2. Molecular Interaction

The molecular structure of the ligands Mygalin, Pam3CSK4, Pam2CSK4 and Zymosan was used in the analysis of the molecular interaction in TLR2/1 and TLR2/6 heterodimers. The molecular interactions of the TLR2/1-Pam3CSK4 and TLR2/6-Pam2CSK4 complexes were directly analyzed in the co-crystallization structure. Molecular docking analysis of Mygalin was carried out on TLR2/1 and TLR2/6 and Zymosan only on TLR2/6. In the docking analysis, only the polypeptide chains of TLR2/1 and TLR2/6 were used; therefore, all accessory molecules, such as water, N-glycosylations, Pam3CSK4, and Pam2CSK4, were removed from the heterodimer receptors, using DSV.

The molecular files of Mygalin and Zymosan were converted to PDBQT format for docking calculations using AutoDock Tools [50]. All rotatable bonds in each ligand remained free, and the amino acids of TLR2/TLR1 and TLR2/TLR6 were kept rigid; hydrogen atoms and Gasteiger charges were assigned. In the docking setup of TLR2/1 of Mygalin, the grid dimensions were x = −18.2, y = −13.8, and z = 9.6 and the size was x = 22, y = 18, and z = 20. In TLR2/6, the grid dimensions for Mygalin were x = 23.6, y = −36.2, and z = 36.4 and the size was x = 22, y = 30, and z = 24; the grid dimensions for Zymosan were x = 21.0, y = −38.0, and z = 36.5 and the size was x = 34, y = 30, and z = 34. Finally, in both heterodimers (TLR2/1 and TLR2/6) the box spacing was 1.0 Å, the exhaustiveness was set to 300, and the molecular docking calculation was performed using AutoDock Vina 1.1.2 [51].

#### 4.2.3. Energy of Interaction of the Receptor–Ligand Complexes

The free energy of the receptor–ligand complexes (TLR2/1 and TLR2/6) was calculated using the coordinates of the co-crystallization and docking calculations, as commented above. In the TLR2/1 and TLR2/6 complexes, the amino acids and molecular interactions that stabilize the receptor–ligand complexes were identified with DSV software (Biovia, 2021). Only the first core of residues belonging to the receptor’s binding pocket was included, and the cut-off distance between the ligands and the receptor was set at 4.5 Å for hydrophobic interactions and 3.4 Å for hydrogen bonds. The interaction energy of the receptor–ligand complexes was calculated. The total free energies of interaction included the ligand molecule and all adjacent binding-pocket residues. A simplified procedure to calculate the interaction energy (IE) was provided by the following formula: IE = [RL] − [R + L], where IE is the interaction energy of binding, RL is the energy of the complex formed by the receptor residues and the ligand, R is the energy of the TLR2/1 or TLR2/6 residues, and L is the energy of the ligand. The binding interaction energies, molecular geometries of ligands, and electronic properties were determined by single-point calculations using Density Functional Theory (DFT) with the ωB97X-D functional and the 6–31 G* basis-set level using Spartan’20 software (for an extensive description of all ab initio methods used in this work, see [52]). Docking and ab initio calculations were performed on a 16-core computer with a Xeon processor at 3.4 GHz.

## 5. Conclusions

The results obtained both in silico and in vitro, using phagocytic cells activated by different microbial products, indicate the potential of Mygalin in controlling the inflammatory response. Mygalin appears to modulate the production of immune mediators and transcription factors crucial for the innate immune response and the subsequent activation of adaptive immunity. Additional studies are currently in progress.

## Figures and Tables

**Figure 1 ijms-25-10555-f001:**
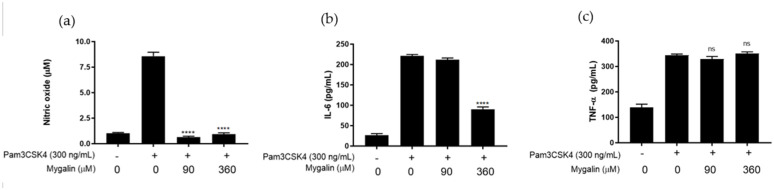
Mygalin interference in the synthesis of immune mediators induced by Pam3CSK4. (**a**) J774A.1 cells (1 × 10^6^ per well) were pre-treated with Mygalin (90 and 360 μM) for 1 h and subsequently activated with Pam3CSK4 (300 ng/mL). Supernatant was collected for (**a**) NO measurement using the Griess method, and (**b**) IL-6 (**c**) TNF-α using ELISA. Data are presented as mean ± SEM of three independent experiments. Data were analyzed using the Student’s *t*-test, comparing Pam3CSK4-stimulated groups treated and untreated with Mygalin **** *p* < 0.0001; ns (not significant).

**Figure 2 ijms-25-10555-f002:**
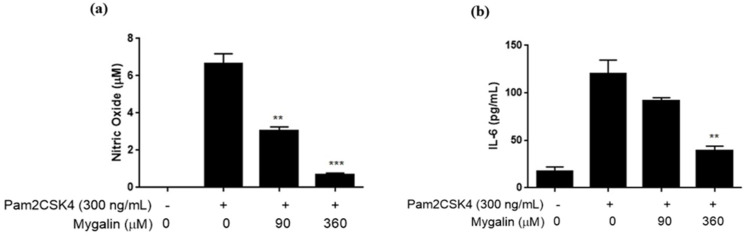
Mygalin interference in the synthesis of immune mediators induced by Pam2CSK4. J774A.1 cells (1 × 10 × 6 per well) were pre-treated with Mygalin (90 and 360 μM) for 1 h and subsequently activated with Pam2CSK4 (300 ng/mL). After 20–24 h, the supernatant was collected for NO measurement using the Griess method (**a**) and IL-6 (**b**) using ELISA. Data are presented as mean ± SEM of three independent experiments. Data were analyzed using the Student’s *t*-test, comparing Pam2CSK4-stimulated groups treated and untreated with Mygalin ** *p* < 0.01/*** *p* < 0.001.

**Figure 3 ijms-25-10555-f003:**
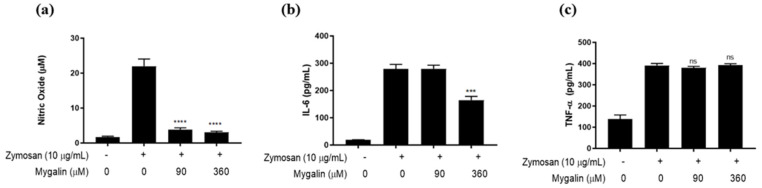
Effect of pre-treatment with Mygalin on the synthesis of immune mediators induced by Zymosan. J774A.1 cells (1 × 10^6^ per well) were pre-treated with Mygalin (90 and 360 μM) for 1 h and subsequently activated with Zymosan (10 µg/mL). After 20–24 h, the supernatant was collected for NO measurement using the Griess method (**a**), and IL-6 (**b**) and TNF-α (**c**) using ELISA. Data are presented as mean ± SEM of three independent experiments. Data were analyzed using the Student’s *t*-test, comparing Zymosan-stimulated groups treated and untreated with Mygalin *** *p* < 0.001/**** *p* < 0.0001; ns (not significant).

**Figure 4 ijms-25-10555-f004:**
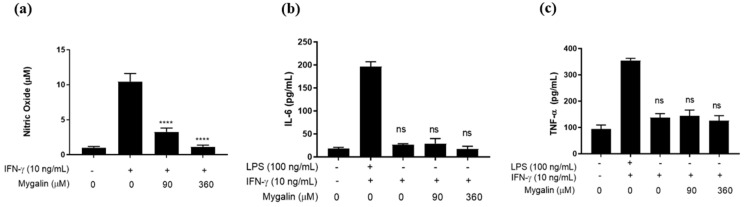
Effect of pre-treatment with Mygalin on IFN-γ-activated J774 macrophages. J774A.1 cells (1 × 10^6^ per well) were pre-treated with Mygalin (90 and 360 μM) for 1 h and subsequently activated with IFN-γ (10 ng/mL). After 20–24 h, the supernatant was collected for NO measurement using the Griess method (**a**), and IL-6 (**b**) and TNF-α (**c**) using ELISA. Data are presented as mean ± SEM of three independent experiments. Data were analyzed using the Student’s *t*-test, comparing IFN-γ-stimulated groups treated and untreated with Mygalin **** *p* < 0.0001; ns (not significant).

**Figure 5 ijms-25-10555-f005:**
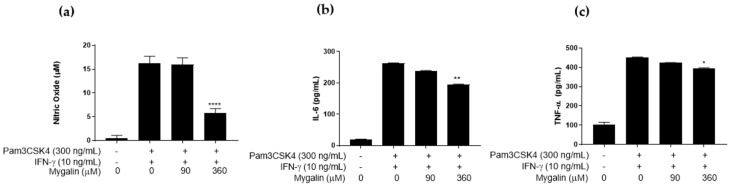
Effect of pre-treatment with Mygalin on J774 macrophages activated by Pam3CSK4 and IFN-γ. J774A.1 cells (1 × 10^6^ per well) were pre-treated with Mygalin (90 and 360 μM) for 1 h and subsequently activated with IFN-γ (10 ng/mL) and Pam3CSK4 (300 ng/mL). After 20–24 h, the supernatant was collected for NO measurement using the Griess method (**a**), and IL-6 (**b**) and TNF-α (**c**) using ELISA. Data are presented as mean ± SEM of three independent experiments. Data were analyzed using the Student’s *t*-test, comparing IFN-γ-stimulated groups treated and untreated with Mygalin * *p* < 0.05/** *p* < 0.01/**** *p* < 0.0001.

**Figure 6 ijms-25-10555-f006:**
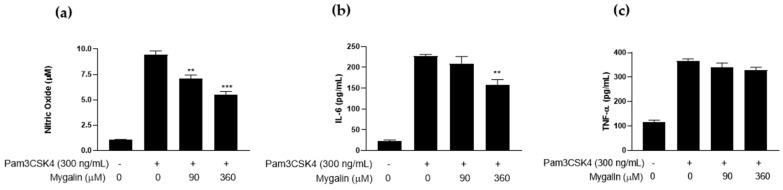
Action of Mygalin on the synthesis of immune mediators in Pam3CSK4-pre-activated macrophages. J774A.1 cells (1 × 10^6^ per well) were pre-activated with Pam3CSK4 (300 ng/mL) for 20 h and subsequently treated with Mygalin (90 and 360 μM). After 20–24 h, the supernatant was collected for NO measurement using the Griess method (**a**), and IL-6 (**b**) and TNF-α (**c**) using ELISA. Data are presented as mean ± SEM of three independent experiments. Data were analyzed using the Student’s *t*-test, comparing groups stimulated with IFN-γ, treated and untreated with Mygalin ** *p* < 0.01/*** *p* < 0.001.

**Figure 7 ijms-25-10555-f007:**
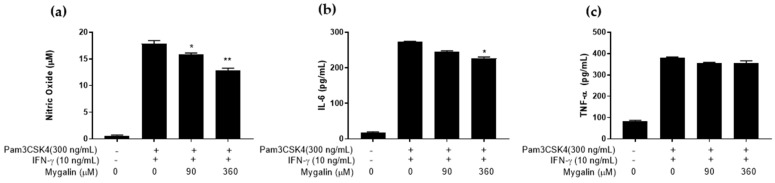
Action of Mygalin on the synthesis of immune mediators in macrophages pre-activated with Pam3CSK4 + IFN-γ. J774A.1 cells (1 × 10^6^ per well) were pre-activated with Pam3CSK4 (300 ng/mL) + IFN-γ (10 ng/mL) for 20 h and subsequently treated with Mygalin (90 and 360 μM). After 20–24 h, the supernatant was collected for NO measurement using the Griess method (**a**), and IL-6 (**b**) and TNF-α (**c**) using ELISA. Data are presented as mean ± SEM of three independent experiments. Data were analyzed using the Student’s *t*-test, comparing groups stimulated with IFN-γ, treated and untreated with Mygalin * *p* < 0.05/** *p* < 0.01.

**Figure 8 ijms-25-10555-f008:**
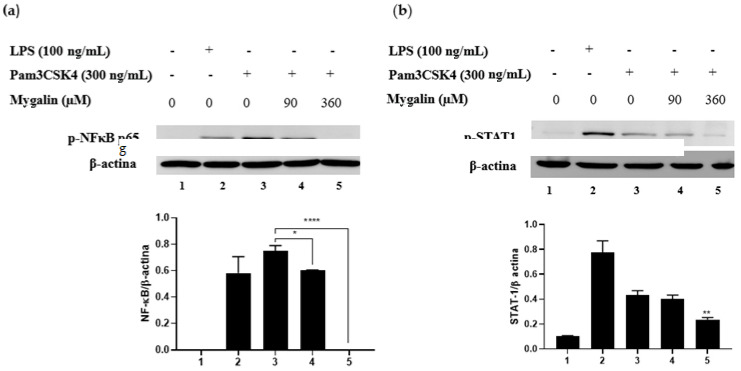
Mygalin suppresses NF-κB p65 and STAT-1 phosphorylation induced by Pam3CSK4. Effect of Mygalin (0–360 µM) on the expression of (**a**) NF-κB p65 and (**b**) STAT1 visualized by Western Blotting using specific antibodies against NF-κB p65 and STAT1. LPS was used as a positive control and β-actin as an internal control. The products were quantified by densitometry and levels of (**a**) NF-κB p65 and (**b**) STAT1 were normalized to β-actin and compared with the Pam3CSK4-treated group. Bars represent mean ± SEM of three independent experiments. (* *p* < 0.05, ** *p* < 0.01, **** *p* < 0.0001).

**Figure 9 ijms-25-10555-f009:**
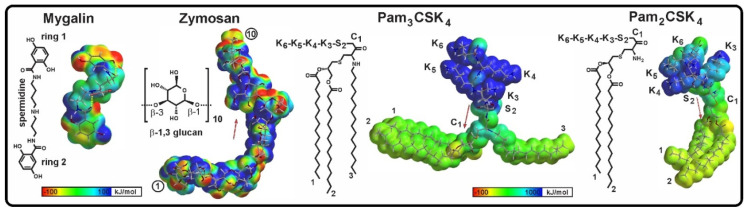
Schematic (**left**) and electronic structure of ligands (**right**). In Pam3CSK4 and Pam2CSK4: residues K (lys), S (ser), C (cys) and the schematic structure in Zymosan represent ten glucopyranosyl units in β-1,3 linkages. Electrostatic potential (EP) and dipole moments vector (red arrows) are shown. Mygalin and Zymosan show heterogeneous EP patterns, with mixed positive and negative spots. Pam3CSK4 and Pam2CSK4 are protonated molecules, which produce EP with two well-defined phases; positive (blue) due to the ammonium group (NH_3_^+^) and neutral (green) due to the lipid chains and, therefore, a large dipole moment oriented from the positive to the neutral region.

**Figure 10 ijms-25-10555-f010:**
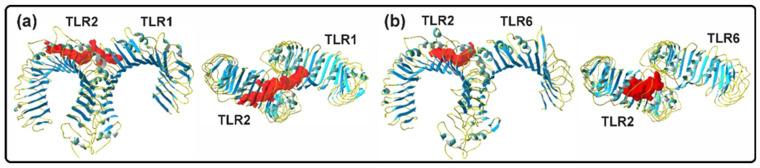
The ligand-binding domain (red) of the TLR2/1 and TLR2/6 heterodimers are shown in front and top view. (**a**) In TLR2/1, a ligand-binding channel is connecting both binding pockets of TLR2 and TLR1. (**b**) In the TLR2/6 dimer, the ligand-binding pocket is only located in TLR2.

**Figure 11 ijms-25-10555-f011:**
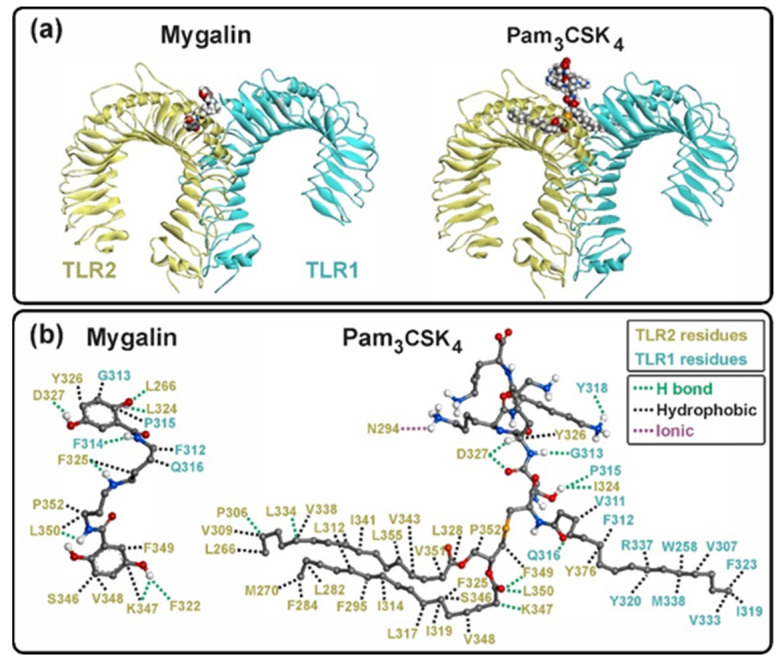
TLR2/1–ligand complex. (**a**) Mygalin and Pam3CSK4 in van der Waals model are shown in complex with TLR2/1. Mygalin occupies mainly the binding pocket of TLR2 and some solvent-exposed residues on the surface of TLR1. Pam3CSK4 occupies both binding pockets in the TLR2/1 dimer. (**b**) Residues of TLR2/1 involved in the interaction with ligands are shown. Hydrophobic (black), hydrogen (green) and ionic (purple) bonding interactions are shown with dotted lines.

**Figure 12 ijms-25-10555-f012:**
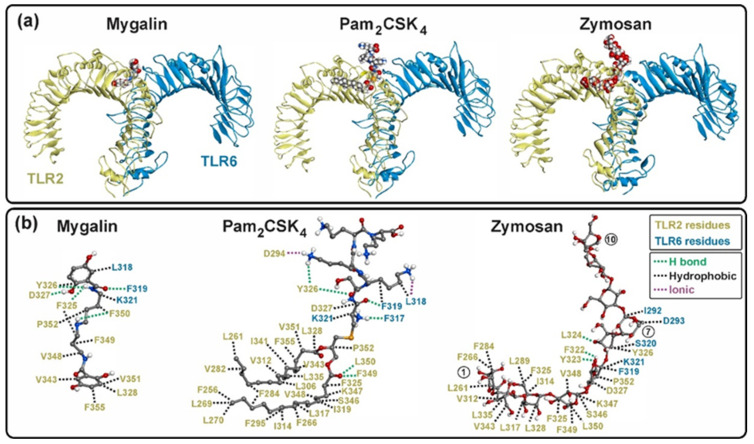
TLR2/6–ligand complex. (**a**) Mygalin, Pam3CSK4 and Zymosan in an der Waals model are shown in complex with TLR2/6. Mygalin occupies mainly the binding pocket of TLR2 and some solvent-exposed residues on the surface of TLR6. Pam3CSK4 and Zymosan occupy deeply the binding pocket in the TLR2 and some residues on the surface of TLR6. (**b**) Residues of TLR2/6 involved in the interaction with ligands are shown. Hydrophobic (black), hydrogen (green) and ionic (purple) bonding interactions are shown with dotted lines. In Zymosan, three monosaccharides are indicated in the circles (1, 7 and 10).

**Table 1 ijms-25-10555-t001:** Molecular properties of the ligands and volume of the ligand-binding domain of TLR2/1 and TLR2/6.

Ligand	Total Energy (AU)	Dipole (Debyes)	Volume (Å^3^)	Polar Surface Area (Å^2^)
Mygalin *	−1431.61883	7.24	414.15	125.99
Mygalin **	−1431.61298	4.46	418.04	135.67
Pam3CSK	−5104.04359	182.55	1685.50	348.53
Pam2CSK4 **	−4401.09779	168.21	1398.31	342.89
Zymosan	−6181.73189	24.03	1390.29	658.13

* TLR2/1 and ** TLR2/6 conformation complex.

**Table 2 ijms-25-10555-t002:** Molecular properties of the volume of the ligand-binding domain of TLR2/1 and TLR2/6.

Ligand-Binding Pocket Volume (Å^3^)		Ligand-Binding Channel Volume (Å^3^)
TLR2	TLR1	TLR2/TLR1
912	255	1161
TLR2	TLR6	TLR2/TLR6
1175	----	1175

## Data Availability

Data are contained within the article.

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
