# Peer review of "Impact of Mygalin on Inflammatory Response Induced by Toll-like Receptor 2 Agonists and IFN-γ Activation"

_ijms, 2024, doi:10.3390/ijms251910555_

Round 1

Reviewer 1 Report (Previous Reviewer 1)

Comments and Suggestions for Authors

The authors have well addressed on all raised issues.

Author Response

Reviewer comments: There were no questions.

Thank you for your valuable assistance in reviewing this manuscript. My email address and the keyword Mygalin were included to the manuscript.

Reviewer 2 Report (New Reviewer)

Comments and Suggestions for Authors

The authors demonstrated that mygalin effectively counteracts established inflammatory responses. Specifically, they showed that pre-treatment with mygalin, followed by activation with TLR2/1 and TLR2/6 agonists, significantly reduced IL-6 synthesis and nitric oxide (NO) production. These effects were also observed in cell lines pre-activated with Pam3CSK4. Computational analysis further revealed that mygalin forms stable complexes at the interface between the TLR dimers, mimicking the transition from the hydrophobic to polar regions found in the agonists. However, a font change (bold) is needed on lines 221–222 on page 6.

Author Response

Reviewer comments: a font change (bold) is needed on lines 221–222 on page 6.

Response: Only one comment was made, but I was unable to find the correction that needed to be made on page 6, lines 221 and 222. My email address and the keyword Mygalin were included in the manuscript.

This manuscript is a resubmission of an earlier submission. The following is a list of the peer review reports and author responses from that submission.

Round 1

Reviewer 1 Report

Comments and Suggestions for Authors

Monamaris et al. submitted the manuscript entitled: Impact of Mygalin on inflammatory response induced by Toll-like receptor 2 agonists and IFN-γ activation, in which the authors focused on the negative regulation effects of Mygalin on the secretion of inflammatory-related  factors. The authors first separately checked the effects of Mygalin in the presence of TLR1/2, agonist, TLR2/6 agonist and IFN-gamma, correspondingly. Next, the authors also compared the aforementioned results with pre-activation of these agonists/factor. Lastly, the authors also checked phosphorylation level of NFkB and STAT1 to further verify TLR activation and regulation effects of Mygalin. While this topic may of interest to potential readers of IJMS, the manuscript contained pretty confusing part. Further modifications are needed.

My comments are as follows.

1.       Page 2, line 53-55: I noticed that the same group published another paper (DOI: 10.3390/biom10121624 ) that Mygalin can also serve as TLR4 agonist. And from my understanding to this work, the authors did not prove the regulation effect came from on-target TLR2 efficacy.

2.       Page 2, figure 1 and Page 14, section 4.1.3: Please determine the catalog number of assay kit used. A minor concern here is whether IL-6 and TNF-alpha concentration reached upper limit.

3.       Page 3, figure 2a: a) Please check if the data of background NO secretion is correct; b) Why the authors did not test TNF-alpha in this section?

4.       Page 2-3, section 2.1.2 and 2.1.3: Why did the authors tested 2 TLR2/6 agonists? Are there any differences in mechanism or phenotype between them?

5.       Page 4, section 2.1.5: a) I understand IFN-gamma stimulation is a common setup in inflammatory cell models. But this seemed to break this manuscript in two parts, TLR agonism and IFN-gamma stimulation. I suggest including a general introduction on potential relations between the two parts, such as the function and secretion of IFN-gamma regulated by TLR2. b) IFN-gamma stimulation will induce macrophage polarization and differentiation. Will these have impact on secretion of these immune mediators?

6.       The authors did observe some differences in phenotype when pre-activated with TLR1/2, agonist TLR2/6 agonist and IFN-gamma. It is suggested to discuss on the findings and explore the potential mechanism in it.

Reviewer 2 Report

Comments and Suggestions for Authors

The authors submitted as the title “Impact of Mygalin on inflammatory response induced by Toll- 2 like receptor 2 agonists and IFN-γ activation”. This manuscript provides experimental data utilizing in silico and in vitro research on the effect of Mygalin on TLR2 inflammatory response. Although the authors demonstrated promising in silico findings, numerous additional mechanistic experiments are predicted to be required for in vitro studies.

Abstract: it should be well-structured, succinct, and detailed to ensure that readers comprehend. There is too much background information presented here (lines 14-27). In contrast, the authors' findings are presented in around four lines. I believe that this point must be replaced.

Introduction: The introduction should offer an abundance of background information that the writers should have before to reading this essay. The authors is focusing on the their point, and most phrases need more evidence (references). This section should be changed and enhanced mostly.

Results and discussion: The authors used various small molecules to investigate the effects of Mygalin on TLR2-mediated inflammatory responses, and they observed that Mygalin reduced the expression of several inflammatory cytokines and NF-kB/STAT1 protein.

It is well established that TLR4 constitute a signaling complex that reacts to the lipopolysaccharide (LPS) of many Gram-negative bacteria. TLR2 recognizes a variety of bacterial products, including components of Gram-positive bacterial cell walls, peptidoglycan, lipoproteins, and lipoteichoic acid. Furthermore, TLR2/4 triggers the inflammatory response by regulating MAPK, NF-KB, and AP1 as underlying mechanisms. In a previous study conducted by the authors, they uncovered the regulation of the TLR4 inflammatory response of Mygalin, which might be believed to have some overlap with the findings of this study. Thus, for insight into the TLR2 inflammatory response, it is imperative to carry out a focused pathway study. Furthermore, the authors examined the inflammatory response of Mygalin even in cells that were exposed to LPS and IFN-g. Regarding IFN-g, it is understood that T cells and NK cells typically produce and release it. It is believed that the importance of this should be emphasized in the disscussion part.

Comments on the Quality of English Language

The language needs to be revised a lot and reviewed by native speakers.